# Cooperative Path Planning for Multiple Mobile Robots via HAFSA and an Expansion Logic Strategy

**Yiqing Huang \*** , **Zhikun Li, Yan Jiang and Lu Cheng**

College of Electrical Engineering, Anhui Polytechnic University, Wuhu 241000, China; lzk52170@163.com (Z.L.); Jiangyan117@163.com (Y.J.); chenglu073@163.com (L.C.)

**\*** Correspondence: yiqhuang@ahpu.edu.cn



**Featured Application: In various unknown environments, cooperative path planning problem of multiple mobile robots is becoming more and more important. The efficiency and reliability can be greatly improved by the cooperation of multiple mobile robots. The novel obstacle avoidance and real-time navigation algorithm presented in this article may be useful for marine exploration, military, aerospace and mining detection. Also, for the developed real-time navigation algorithm, the presented hybrid artificial fish swarm algorithm and expansion logic strategy are helpful not only for accelerating the convergence rate, but also for improving decision-making ability.**

**Abstract:** The cooperative path planning problem of multiple mobile robots in an unknown indoor environment is considered in this article. We presented a novel obstacle avoidance and real-time navigation algorithm. The proposed approach consisted of global path planning and local path planning via HAFSA (hybrid artificial fish swarm algorithm) and an expansion logic strategy. Meanwhile, a kind of scoring function was developed, which shortened the time of local path planning and improved the decision-making ability of the path planning algorithm. Finally, using STDR (simple two dimensional robot simulator) and RVIZ (robot operating system visualizer), a multiple mobile robot simulation platform was designed to verify the presented real-time navigation algorithm. Simulation experiments were performed to validate the effectiveness of the proposed path planning method for multiple mobile robots.

**Keywords:** path planning; multiple mobile robots; artificial fish swarm algorithm; expansion logic strategy

## 1. Introduction

Mobile robots can be equipped with different sensors and tools to afford a variety of services such as home care, mining detection and object handling [1,2]. One of the fundamental issues with mobile robots performing tasks is ensuring that they can navigate safely in an unknown indoor environment. Therefore, path planning is crucial for the successful application of mobile robots. The goal of mobile robot path planning is to find a motion path from a starting position to a target position in an environment with obstacles [3–5]. For the past two decades, there has been a great deal of research on the problem of mobile robot path planning. For example, a novel motion map was constructed for mobile robots based on the BIE (boundary integral equation) method, and then, a point-to point path planning problem was addressed in a known environment with static obstacles [6]. Furthermore, an improved three-dimensional-like grid map was developed to represent the environment model [7], and then, a simple but efficient path planning algorithm was presented to solve robot navigation problems in a static environment. The authors designed an autonomous multi-goal navigation system for picking up or delivering tasks in mobile robotics and a multi-goal path planning method based on

the Lin−Kernighan heuristics (LKH) algorithm for intelligent service mobile robots in Reference [8]. There are also some intelligent methods that can be applied to mobile robot path planning. The cross probability and the mutation probability for GA (genetic algorithm) were improved and the improved algorithm was applied to the path planning problem of mobile robots in Reference [9], whereas the authors proposed an intelligent motion planning and navigation method for omnidirectional mobile robots via a fuzzy logic algorithm in Reference [10]. Moreover, the developed navigation system is especially suitable for real-time path planning applications. A novel optimal hierarchical global path planning method for mobile robots in a cluttered environment was presented in Reference [11]. In this method, a combination of the triangular decomposition approach, constrained multi-objective PSO (particle swarm optimization) and Dijkstra's algorithm is presented in order to obtain an optimal path planning trajectory. In addition, due to good feedback information and better distributed computing, the authors proposed a path planning method for mobile robots via an improved ant colony algorithm in grid maps in Reference [12].

Using multiple mobile robots rather than a single mobile robot can improve working capability and performance. Therefore, recently, research on multiple mobile robots has become a hot topic. In previous studies, only static obstacles in the unknown environment were considered. For holonomic wheeled mobile robots in static environments, an optimal multiple mobile robot path planning method based on adaptive charged system search (CSS) algorithms was addressed [13]. However, path planning methods in an unknown environment with dynamic obstacles are even more acute in multiple mobile robot areas. In Reference [14], a new path planning approach for coordinating multiple mobile robots was presented and the authors developed an online strategy to adjust path planning for avoidance of dynamic obstacles. Furthermore, a biologically inspired neural-network-based intelligent method was proposed for a multiple robot system with moving obstacles [15]. The proposed method could plan the paths of multiple robots to avoid collision with dynamic obstacles.

Although these previously-developed navigation algorithms have shown good performance for solving robot path planning, they have also shown some limitations such as slow convergence and a local optimum. The local optimal problem is the most common problem in solving path planning. Motivated by the aforementioned reasons, we reconstructed an analytical real-time cooperative navigation algorithm to accommodate multiple mobile robot systems. The proposed EAFSA (empirical artificial fish swarm algorithm) was used to avoid falling into local optimal problems and to realize global path planning for a single mobile robot. Then, an expansion logic strategy was introduced to avoid collisions between multiple mobile robots and an environment with obstacles. A multiple mobile robot simulation system was developed using STDR (simple two dimensional robot simulator) and RVIZ (robot operating system visualizer) software. Finally, the presented method was proven to be effective by experiments conducted in a simulated environment.

The main contributions of the article are summarized as follows. (1) EAFSA is presented to solve the global path planning problem for a single mobile robot; (2) an expansion logic strategy and a kind of scoring function are proposed for a multiple mobile robot real-time navigation algorithm. The presented real-time navigation algorithm is helpful not only for accelerating the convergence rate, but also for improving decision-making ability.

The remainder of this paper is organized as follows: Section 2 describes the process of the presented HAFSA. Section 3 presents the developed expansion logic strategy and scoring function. Section 4 shows the results of a simulation to demonstrate the performance of the proposed algorithms. The concluding remarks are given in Section 5.

## 2. Hybrid Artificial Fish Swarm Algorithm

The artificial fish swarm algorithm (AFSA) is a novel swarm intelligent optimization method inspired by natural fish swarm behavior. It has been successfully used in the field of wireless telemedicine systems [16], fault diagnosis [17], indoor visible light positioning [18], floating wind turbines [19], etc.

The basic idea of AFSA can be described as follows: If the position of each artificial fish is $X = (x_1, x_2, \cdots, x_n)$ and the size of the fish population is *Num*, $Y$ denotes the food concentration of the artificial fish in the current position and $Y = f(X)$ is fitness or the objective function at position $X$. Each artificial fish tries to find an optimal position to satisfy their food needs using preying behavior, swarming behavior, following behavior and random behavior [20].

**(1)  Preying Behavior**

If the current state of an artificial fish is $X_i(t)$, $X_j(t)$ is the random state of its visual distance, and $X_{i+1}(t)$ is the next position of $X_i(t)$. If food concentration is $Y_i < Y_j$, the artificial fish swims a *step* in the direction of $X_j(t)$. Otherwise, it randomly selects a state again and judges whether it satisfies the aforementioned condition. In other words, preying behavior can be expressed by the following equation:

$$X_{i+1}(t) = \begin{cases} X_i(t) + \frac{X_j(t) - X_i(t)}{\|X_j(t) - X_i(t)\|} \times step \times rand(0,1) & if \ Y_i < Y_j \\ X_i(t) + Visual \times rand(0,1) & if \ Y_i \geq Y \end{cases}. \tag{1}$$

**(2)  Swarming Behavior**

When $N_F$ is the number of artificial fishes in the current position $X_i(t)$, $X_c(t)$ is the center position of the artificial fishes in their current neighborhood. $if \ Y_c/N_F > \delta Y_i$ is satisfied, the artificial fish moves to a center position, according to Equation (2), due to high food concentration and to avoid crowding each other. Otherwise, the artificial fish executes preying behavior.

$$X_i(t) + \frac{X_c(t) - X_i(t)}{\|X_c(t) - X_i(t)\|} \times step \times rand(0,1) \quad if \ Y_c/N_F > \delta Y_i \tag{2}$$

**(3)  Following Behavior**

Let $X_{\max}(t)$ be the local best companion with food concentration $Y_{\max}$ in the current neighborhood of $X_i(t)$. $if \ Y_{\max}/N_F > \delta Y_i$ is satisfied, the artificial fish moves to a position according to Equation (3). Otherwise, the next position of the artificial fish can be obtained by preying behavior.

$$X_i(t) + \frac{X_{\max}(t) - X_i(t)}{\|X_{\max}(t) - X_i(t)\|} \times step \times rand(0,1) \quad if \ Y_{\max}/N_F > \delta Y_i \tag{3}$$

**(4)  Random Behavior**

The artificial fish chooses an arbitrary state or position randomly in its *Visual* field, and then it swims towards the selected state. Random behavior is a default behavior and it can be described as

$$X_i(t) + Visual \times rand(0,1), \tag{4}$$

where $X_i(t)$ is the current state of the artificial fish and $X_{i+1}(t)$ is the next position of $X_i(t)$.

Given the above consideration, an effective hybrid fish swarm algorithm (HFSA) with experiential learning and a detection operator was presented to solve the local optimal problem and realize global path planning for a single mobile robot.

*2.1. Experiential Learning*

In this section, an experiential learning strategy is presented to improve the performance of AFSA. Experiential learning strategies include adjustment of the step size and food concentration for the artificial fish. The step size of an artificial fish is fixed in traditional AFSA. However, as is known, the step size determines the convergence rate. If the step size is too small, the artificial fish will reach the

optimal solution slowly and the global search ability will be decreased. Therefore, the artificial fish is easy to fall into a local optimum. Conversely, if the step size is too large, the convergence speed will be increased and oscillation will occur later in the algorithm iteration. Therefore, it is necessary to select an appropriate step size to ensure the global convergence speed and improve the accuracy of the optimal solution. In this article, a logarithmic function was used to update the step size by Equation (5);

$$b = \log_N p, p = 1, 2, 3 \cdots N, \tag{5}$$

where $p$ indicates the current iteration number, and $N$ is the maximum number of iterations.

Then, the step size in the population update formula could be obtained by

$$step = \log_N p \cdot (X_\varepsilon(t) - X_i(t)) p = 1, 2, 3 \cdots N, i = 1, 2, 3 \cdots NUM, \tag{6}$$

where $X_i(t)$ is the current state of the artificial fish, $X_{i+1}(t)$ is the next position of $X_i(t)$ and $X_\varepsilon(t)$ is the state that needs to be searched.

Finally, the position of the updated solution could be expressed as follows:

$$X_{i+1}(t) = X_i(t) + \log_{NC} p \cdot (X_\varepsilon(t) - X_i(t)) p = 1, 2, 3 \cdots N, i = 1, 2, 3 \cdots NUM. \tag{7}$$

As shown in Equation (7), with the increase in the number of iterations, the moving step gradually adapts to the change of the iteration numbers. The local search ability is increased by the improved moving factor. As a result, the artificial fish can locate the search direction quickly, move to the target area, maintain the global search ability of the optimal solution and accelerate the convergence speed.

On the other hand, the food concentration of the artificial fish was represented as the ability of the solution to solve an optimization problem. The current position of the artificial fish with the highest food concentration was the optimal solution of the optimization problem. In this article, a weight coefficient function was used to design the food concentration, which could decrease the food concentration of the artificial fish and avoid the problem that the suboptimal solution of the fish swarm algorithm would interfere with the global solution. The food concentration equation was designed as follows:

$$H_{\exp} = w\sqrt{(x_i - x_g)^2 + (y_i - y_g)^2}, \tag{8}$$

where the weight coefficient is $w \in [1, 1.5)$.

### 2.2. Detection Operator

A detection operator was developed to optimize the resulting path trajectory. If $p(kx, ky)$ is the position coordinate of the optimal fish group solved at the $k$th time, $n$ is the dimension of the grid-based map and $k_t$ is the number of the solutions that have been optimized. $goal(x, y)$ is the position coordinate of the target point. The detection operator $R(t)$ can be described as follows:

$$R(t) = \begin{cases} \lceil D(k)/(2k_t) \rceil \times \sqrt{2} & 0 < t < count \\ 2\sqrt{2} & t \geq count \end{cases} \tag{9}$$

$$D(k) = \frac{\|p(kx, ky) - goal(x, y)\|}{\sqrt{2}} \tag{10}$$

$$count = \lfloor [0.27n + 0.5] \times 75\% \rfloor, \tag{11}$$

where $D(k) = \frac{\|p(kx, ky) - goal(x, y)\|}{\sqrt{2}}$, $count = \lfloor [0.27n + 0.5] * 75\% \rfloor$, $\lfloor \cdot \rfloor$ and $\lceil \cdot \rceil$ round toward negative or positive infinity.

The outline of the presented algorithm is described in the following steps:

Step 1.  Initialize the population size *NUM*, the parameters *step* and *visual*, and the maximum number of iterations *N*.

Step 2.  Update the step size and position of the artificial fish using Equation (7).

Step 3.  Calculate the food concentration for each artificial fish using Equation (8) and record the optimal value in the bulletin board.

Step 4.  Perform preying behavior, swarming behavior, following behavior and random behavior.

Step 5.  Check the termination condition. If the stopping condition is satisfied, terminate the iteration process and output optimal solution. Otherwise, return to Step 2.

## 3. Local Path Planning Based on an Expansion Logic Strategy

In this section, an expansion logic strategy is presented to avoid collisions between multiple mobile robots and an environment with obstacles. It plays a decisive role in real-time navigation of mobile robots. Figure 1 shows the obstacle information in an unknown environment. For any polygonal obstacle, the minimum circumscribed circle (MCCI) method and wire envelopes method can be used to perform obstacle expansion operations. In this article, as we sought a rapid expansion method, we adopted the endpoint connection method to generate a circular equation instead of the MCCI method. The expansion logic strategy is described as follows:

If the vertices of n-sided polygonal obstacles are denoted as $p_i(x_i, y_i)$, the Euclidean distance $d_{ij}$ between two vertices $p_i(x_i, y_i)$ and $p_j(x_j, y_j)$ can be expressed by the following equation:

$$d_{ij} = \sqrt{(x_i - x_j)^2 + (y_i - y_j)^2}.$$ (12)

Therefore, the diameter of a range circle is given by

$$d_{\max} = \max\{d_{ij}\}, i, j = 1, 2, \cdots, n$$ (13)

and the circle equation is given by Equation (10).

$$(x - \frac{x_i + x_j}{2})^2 + (y - \frac{y_i + y_j}{2})^2 = (\frac{d_{\max}}{2})^2$$ (14)

In this paper, an environmental map with n-sided polygonal obstacles is shown in Figure 1a. Following the circle Equation (14), two concentric circles with diameters $1.2d_{\max}$ and $1.4d_{\max}$ could be obtained (Figure 1b). As the figure shows, the collision probability was 0.85 for the complementary set area of the intersection between the first circle and the polygonal obstacles. Furthermore, the collision probabilities were 0.65 and 0.45 when the robots moved in the other two grid areas.

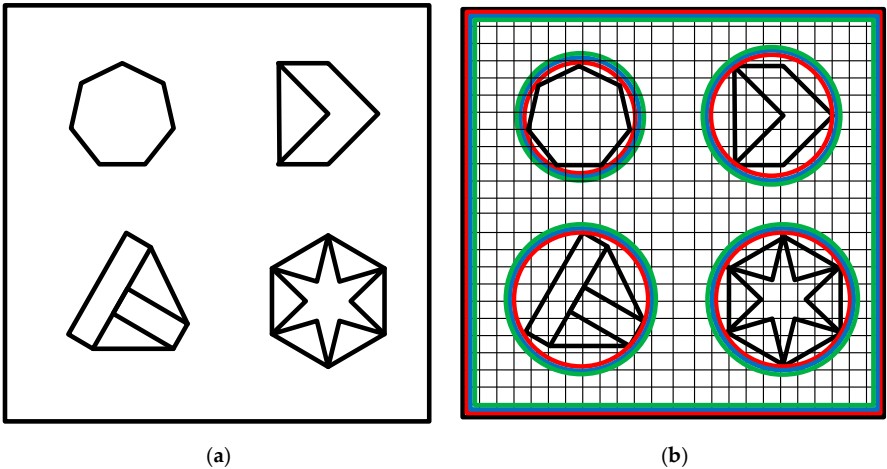

(a)                                           (b)

**Figure 1.** (**a**) Environmental map information; (**b**) expansion logic operation for obstacles.

To evaluate the grid-based environment map, a kind of scoring function is given by Equation (15), which shortens the time of local path planning and improves decision-making ability.

$$score = 100 - \lceil \frac{dist}{l} \rceil, \tag{15}$$

where *dist* denotes the Euclidean distance between the starting point and the target point, and *l* represents the Euclidean distance between the starting point and the current position of the robot.

From the vertical line of the motion direction, we scored the surrounding grids by Equation (15) and the obtained grid scores are shown in Figure 2a. Then, in accordance with the current position information and Equation (12), the mobile robot selected the grid that was the closest to the target point as the position of the next moment. If the distance between the grids was equal, preference was given to the high score grid. Finally, the local path planning trajectory, which is indicated by a dotted line, could be obtained using the expansion logic strategy (Figure 2b).

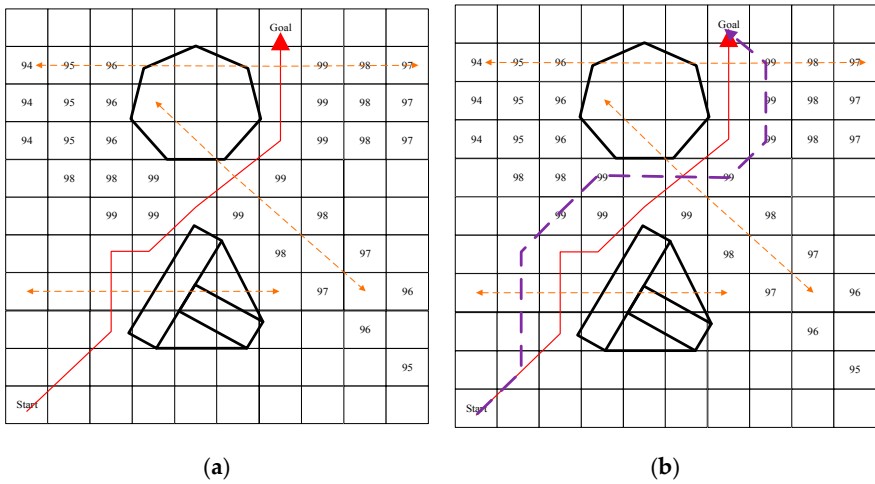

(**a**)     (**b**)

**Figure 2.** (**a**) Local path planning and grid scores; (**b**) local path planning using the expansion logic strategy.

## 4. Simulation Experiments

In this section, to verify the superiority of the presented algorithm for a single mobile robot and $20 \times 20$ grid-based environment maps, the path planning results under the presented hybrid artificial fish swarm algorithm and the traditional fish swarm algorithm are shown in Figure 3.

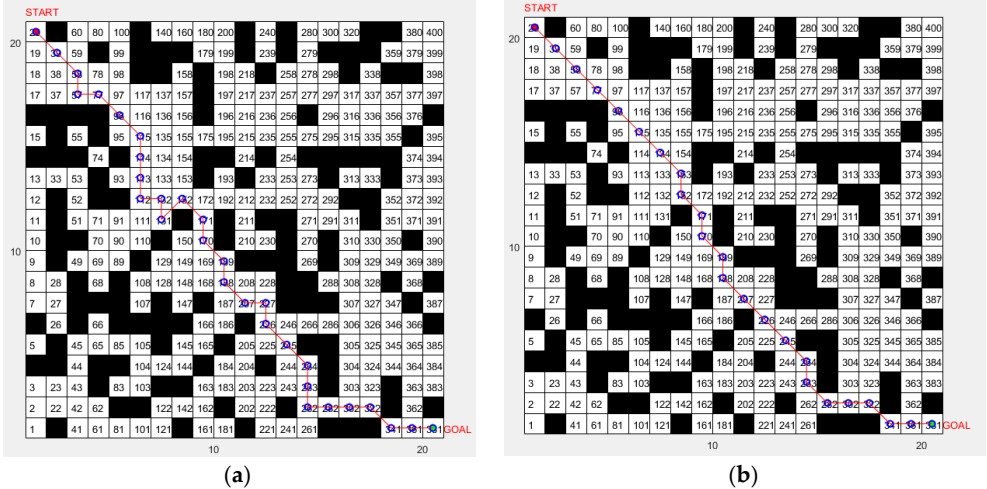

(**a**)     (**b**)

**Figure 3.** (**a**) Artificial fish swarm algorithm; (**b**) the presented hybrid artificial fish swarm algorithm.

For 20 × 20 and 40 × 40 grid-based environment maps, the path planning results based on the adaptive artificial fish swarm algorithm (AAFA) [21], fuzzy logic (FL) [22], the improved genetic algorithm (IGA) [23] and our method are shown in Table 1. As can be seen in Figure 3 and Table 1, optimization performance and iteration time can be improved by using the presented method.

**Table 1.** Performance comparison of four algorithms.

| Environment Map | Algorithms | The Longest Path Length | The Optimal Path Length | The Average Path Length | Iteration Time/s |
|---|---|---|---|---|---|
| 20 × 20 grids | AAFA | 35.1283 | 30.0348 | 33.6231 | 16.5182 |
| | FL | 34.3848 | 29.7990 | 32.0919 | 12.2304 |
| | IGA | 32.3254 | 29.6325 | 30.8652 | 16.1826 |
| | Our method | 30.3848 | 29.2132 | 29.7990 | 9.3102 |
| 40 × 40 grids | AAFA | 80.2372 | 73.7103 | 79.2293 | 98.5621 |
| | FL | 75.1838 | 69.4975 | 72.3407 | 90.4073 |
| | IGA | 66.4723 | 62.9002 | 65.7213 | 97.7652 |
| | Our method | 64.0833 | 61.4264 | 62.7549 | 74.5801 |

Furthermore, simulation results were performed on a group of mobile robots. A multiple mobile robot navigation system, which included three mobile robots labeled as Robot 0, Robot 1 and Robot 2, was designed using STDR (simple two dimensional robot simulator) and RVIZ (robot operating system visualizer) software. Each mobile robot was equipped with four ultrasonic sensors and a radar detector with laser detection capabilities. The environment map and obstacle information of the simulation experiments are shown in Figure 4a. If the size of the empirical fish swarm was $N = 50$, and the visual distance of an artificial fish was $v = 10$, the crowd factor was $\delta = 0.618$.

As shown in Figure 4, the initial poses of the three mobile robots were (1, 1, 0), (1, 5, 0) and (1, 9, 0). The goal positions were (10, 14), (17, 1) and (18, 13).

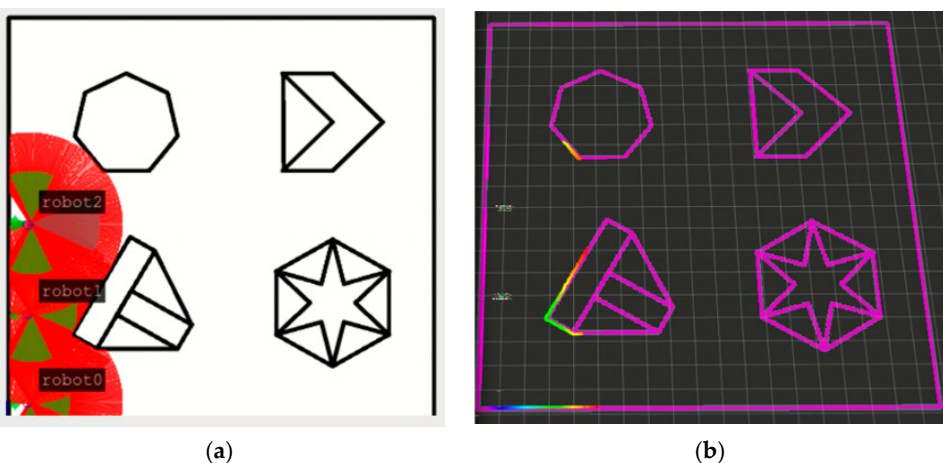

(a)  (b)

**Figure 4.** (**a**) The initial poses of the three mobile robots; (**b**) simulation visual interface of multiple mobile robots.

The global path planning trajectories of the three mobile robots are shown in Figure 5. As shown in Figure 5a, Robot 0 (initial pose was (1, 1, 0)), Robot 1 (initial pose was (1, 5, 0)) and Robot 2 (initial pose was (1, 9, 0)) started to move, and then they updated their motion paths using the expansion logic strategy to avoid mutual collision. Meanwhile, many feasible paths could be obtained by the presented EFSA. Therefore, we can see that the global path planning results for mobile robots were not unique (Figure 5b).

Figure 6a illustrates the position information (the poses of the three mobile robots were (3, 10, 0), (9, 1, 0) and (10, 10, 0)) at a specific time in the simulation experiment. Local and global paths of multiple mobile robots are shown in Figure 6b.

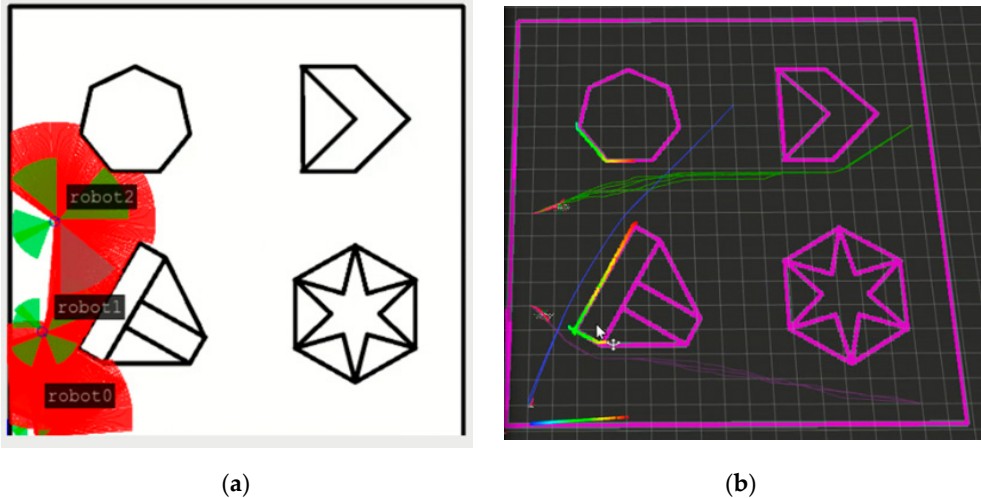

(**a**)　　　　　　　　　　　　　　　　　　　　(**b**)

**Figure 5.** (**a**) The initial poses of the three mobile robots; (**b**) global path planning trajectories.

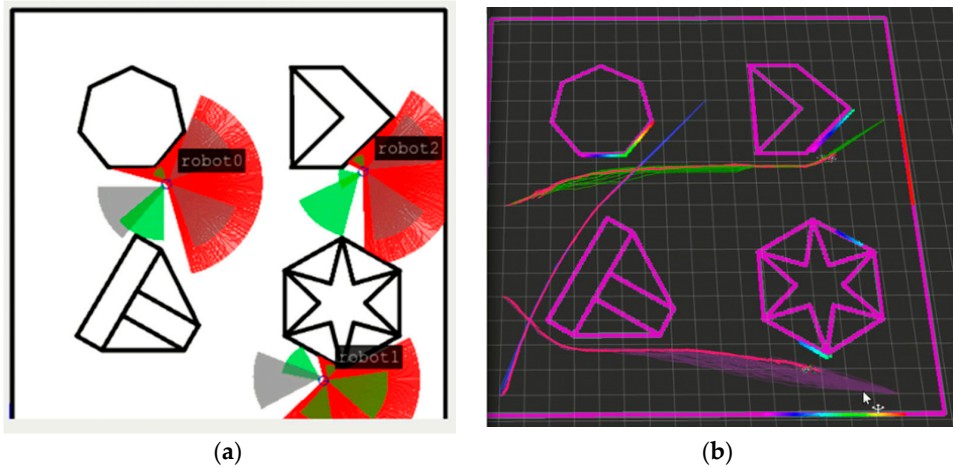

(**a**)　　　　　　　　　　　　　　　　　　　　(**b**)

**Figure 6.** (**a**) Position information at a specific time; (**b**) local and global paths of multiple mobile robots.

The positions of Robot 0, Robot 1 and Robot 2 are shown in Figure 7a when the robots reached the target points. Figure 7b illustrates the final paths of the three mobile robots after reaching the target points.

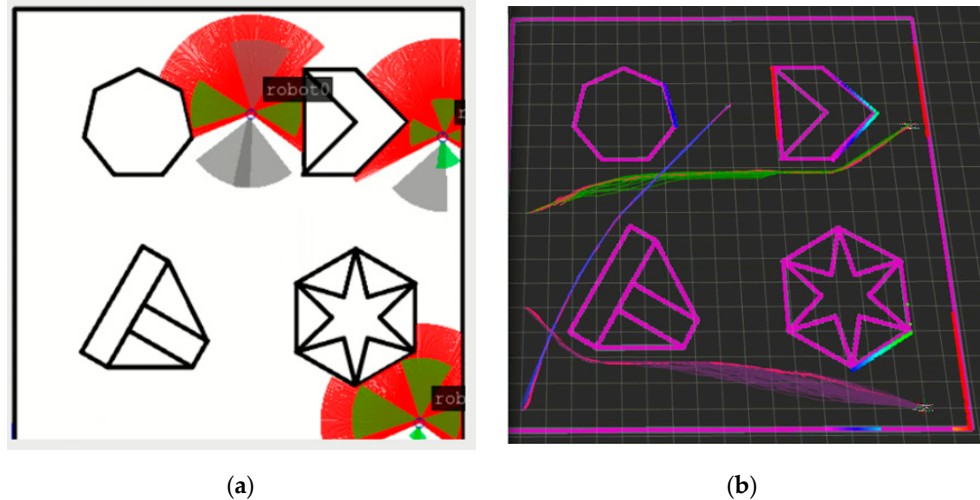

(**a**)　　　　　　　　　　　　　　　　　　　　(**b**)

**Figure 7.** (**a**) The robots reached the target points; (**b**) the final paths of the three mobile robots.

## 5. Conclusions

This article focused on the cooperative path planning problem of multiple mobile robots in an unknown environment with obstacles. HAFSA (hybrid artificial fish swarm algorithm) was proposed to solve the local optimal problem and realize cooperative path planning for multiple mobile robots. An experiential learning strategy was presented to improve the performance of AFSA and a detection operator was developed to optimize the resulting path trajectory. In particular, an expansion logic strategy was used to avoid collision between multiple mobile robots and an environment with obstacles. In order to evaluate a grid-based environment map, a kind of scoring function was designed. Finally, a multiple mobile robot simulation system was developed by utilizing STDR and RVIZ software; simulation experimental results validated the effectiveness of the proposed real-time navigation algorithm.

**Author Contributions:** Validation, Z.L. and Y.J.; formal analysis, L.C.; writing—original draft preparation, Y.H.; supervision, Y.H.; project administration, Y.H.

**Funding:** This work was supported in part by Natural Science Research Key Projects in Universities of Anhui Province (KJ2018A0110), Natural Science Foundation of Anhui Province (1608085MF146) and the Foundation for talented young people of Anhui Polytechnic University (2016BJRC004, 2016BJRC008).

**Conflicts of Interest:** The authors declare no conflict of interest.

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
