# Peer review of "Cooperative Path Planning for Multiple Mobile Robots via HAFSA and an Expansion Logic Strategy"

_applsci, doi:10.3390/app9040672_

Reviewer 1 Report

The concept of swarm control based obstacle avoidance of a mobile robot has been suggested.

I think that the experiment in this paper is not enough to evaluate and discuss about the proposed method that the method has already proposed in a previous study.  And the approach is quite easy to get but reasonable by using simulation, the simulation results are limited.

I think that the paper contains less nouveaute and some unclarities on the contents of the real-time path plan.

Experimental results are needed to confirm real-time cooperative path planning. The paper require a variety of experiments and major revisions to be published as regular paper.

Author Response

Response to Reviewer 1 Comments

Dear Reviewer:

   On behalf of my co-authors, we thank you very much for giving us an opportunity to revise our manuscript; we appreciate editor and reviewers very much for their positive and constructive comments and suggestions on our manuscript. We have studied comments carefully and have made correction which we hope meet with approval. Revised portion are marked in red in the paper. The main corrections in the paper and the responds to the reviewer’s comments are as follows:

Point 1: The concept of swarm control based obstacle avoidance of a mobile robot has been suggested. I think that the experiment in this paper is not enough to evaluate and discuss about the proposed method that the method has already proposed in a previous study.  And the approach is quite easy to get but reasonable by using simulation, the simulation results are limited. I think that the paper contains less nouveaute and some unclarities on the contents of the real-time path plan. Experimental results are needed to confirm real-time cooperative path planning. The paper require a variety of experiments and major revisions to be published as regular paper.

Response 1: Firstly, thanks to reviewer for the thoughtful and thorough review. In the revised manuscript, the authors added the path planning results for a single mobile robot under the presented hybrid artificial fish swarm algorithm and traditional fish swarm algorithm in Figure 3. Different from the existing research results, in our method, expansion logic strategy is used to avoid collisions between multiple mobile robots and an environment with obstacles. Especially, a kind of scoring function is designed to evaluate grid-based environment map, which shortens the time of local path planning and improves the decision-making ability of the presented algorithm. Due to the complexity of multi-robot cooperative path planning, at present, we have only done some work in experimental simulation using simple two dimensional robot simulator and robot operating system visualizer software. We will build a multi-robot path planning experimental platform in real environment in the next step.

Once again, thank you very much for your comments and suggestions.. Hopefully we have addressed all of your concerns.

Reviewer 2 Report

Although the main objective of the paper is foccused on multiple mobile robots I find the experiments/simulations with multi robot experiments are sparse.

It would be recommendable to perform more simulations with different scenarios, different number of robots, etc and define clearly the criteria to evaluate the results. 

The content of sections 4 and 5 is poor. It should be revised. 

Author Response

Response to Reviewer 2 Comments

Dear Reviewer:

   On behalf of my co-authors, we thank you very much for giving us an opportunity to revise our manuscript; we appreciate editor and reviewers very much for their positive and constructive comments and suggestions on our manuscript. We have studied comments carefully and have made correction which we hope meet with approval. Revised portion are marked in red in the paper. The main corrections in the paper and the responds to the reviewer’s comments are as follows:

Point 1: Although the main objective of the paper is foccused on multiple mobile robots I find the experiments/simulations with multi robot experiments are sparse. It would be recommendable to perform more simulations with different scenarios, different number of robots, etc and define clearly the criteria to evaluate the results. The content of sections 4 and 5 is poor. It should be revised. 

Response 1: Firstly, thanks to reviewer for the thoughtful and thorough review. In the revised manuscript, to verify the superiority of the presented algorithm, for a single mobile robot and 20×20 grid-based environment maps, more simulation results are added. For example, the path planning results for a single mobile robot under the presented hybrid artificial fish swarm algorithm and traditional fish swarm algorithm are shown in Figure 3. Also, we have revised sections 4 and 5 in the revised manuscript (Revised portion are marked in red).

Once again, thank you very much for your comments and suggestions.. Hopefully we have addressed all of your concerns.

Round  2

Reviewer 1 Report

I think that the paper was revised and supplemented well according to the review comments.

starting and goal points in Figures 4, 5, and 6 are not clear, you should present it clearly.

and, Figure 7 (a) shows some cuts, which need to be made clear.

Experimental data for 40x40 presented in Table 1 is presented, but simulation results should be presented.

Author Response

Response to Reviewer 1 Comments

Dear Reviewer:

   On behalf of my co-authors, we appreciate editor and reviewers very much for their positive and constructive comments and suggestions on our manuscript. Revised portion are marked in red in the paper. The main corrections in the paper and the responds to the reviewer’s comments are as follows:

Point 1: I think that the paper was revised and supplemented well according to the review comments. Starting and goal points in Figures 4, 5, and 6 are not clear, you should present it clearly. and, Figure 7 (a) shows some cuts, which need to be made clear. Experimental data for 40x40 presented in Table 1 is presented, but simulation results should be presented.

Response 1: Thanks to reviewer for the thoughtful and thorough review. For Figures4-6, we presented starting and goal points in the revised manuscript (marked in red in the paper). Meanwhile, we updated Figure 7(a) in the revised manuscript.

It's important to note that there are 1600 grids for 40×40 grid-based environment map; the path cannot see clearly due to excessively small grids, therefore, we did not present the simulation results. But, we presented experimental data for 40x40 in Table 1.

Once again, thank you very much for your comments and suggestions. Hopefully we have addressed all of your concerns.

Reviewer 2 Report

None

Author Response

Dear Reviewer:

   On behalf of my co-authors, we appreciate editor and reviewers very much for their positive and constructive comments and suggestions on our manuscript

   We have checked the spelling and grammar carefully in the revised manuscript. Once again, thank you very much for your comments and suggestions.